# Oxidant-Based Cytotoxic Agents During Aging: From Disturbed Energy Metabolism to Chronic Inflammation and Disease Progression

**DOI:** 10.3390/biom15040547

**Published:** 2025-04-09

**Authors:** Jürgen Arnhold

**Affiliations:** Institute of Medical Physics and Biophysics, Medical Faculty, Leipzig University, Härtelstr. 16-18, 04107 Leipzig, Germany; juergen.arnhold@medizin.uni-leipzig.de

**Keywords:** aging, cytotoxic agents, antagonizing principles, inflammaging, hypoxia, redox homeostasis, oxidative stress, reactive species

## Abstract

In humans, aging is an inevitable consequence of diminished growth processes after reaching maturity. The high order of biomolecules in cells and tissues is continuously disturbed by numerous physical and chemical destructive impacts. Host-derived oxidant-based cytotoxic agents (reactive species, transition free metal ions, and free heme) contribute considerably to this damage. These agents are under the control of immediately acting antagonizing principles, which are important to ensure cell and tissue homeostasis. In this review, I apply the concept of host-derived cytotoxic agents and their interplay with antagonizing principles to the aging process. During aging, energy metabolism and the supply of tissues with dioxygen and nutrients are increasingly disturbed. In addition, a chronic inflammatory state develops, a condition known as inflammaging. The balance between oxidant-based cytotoxic agents and protective mechanisms is analyzed depending on age-based physiological alterations in ATP production. Disturbances in this balance are associated with the development of age-related diseases and comorbidities. An enhanced production of reactive species from dysfunctional mitochondria, alterations in cellular redox homeostasis, and adaptations to hypoxia are highlighted. Examples of how disturbances between oxidant-based cytotoxic agents and antagonizing principles contribute to the pathogenesis of diseases in persons of advanced age are given.

## 1. Introduction

Aging is accompanied by a gradual decline in physiological functions and physical activity and is associated with an increased predisposition to different health-threatening diseases. Despite the existence of about 300 theories and hypotheses about the reasons for aging, there does not exist any comprehensive description explaining all miraculous facets of the aging process [1]. These theories on aging can be roughly divided into the following three main categories: (i) theories related to deviations in descriptions of genetic information, (ii) theories favoring any problems in hormone metabolism, and (iii) theories based on the accumulation of waste products in cells and organisms [2]. Although numerous facts are provided in favor of all these theories, it remains unknown how these different approaches can be synergized into a unifying theory.

In an attempt to convey a more complex description of the aging process, I propose that aging is an inevitable consequence of growth limitation during the development of complex multicellular organisms [2,3]. On the one hand, in a growing organism, cells permanently divide, thus increasing the mass of cells, tissues, and the extracellular matrix. In order to adequately supply all cells of this organism with energy and food, the differentiation and functional specification of cells and the formation of specific tissue types and organs are mandatory. This further increases the complexity of the developing organism. On the other hand, biomolecules, cell organelles, cells, and tissues have numerous chemical and physical impacts that can disturb the integrity of this highly ordered living matter, thus impairing physiological processes in the growing organism. Despite the existence of numerous immediate-acting and inducible protective mechanisms in living matter, only a sufficiently high growth rate can ensure functionally intact organisms. After an organism reaches its optimum size, the general growth rate declines more and more. As a result, destructive elements cannot be compensated for as efficiently as before [3]. There is also a gradual shift in the energy balance in elderly organisms from organism-specific processes like active movement, sensing the environment, food intake, and physical and mental work towards the predominance of processes counterbalancing any deviations from normal physiological functioning on a systemic and cellular level [2]. With an increasing age, more and more energy equivalents are required to maintain the essential physiological parameters in the cells and tissues of an organism.

In organisms, important protective functions are realized by the immune and its associated systems, like the acute-phase, complement, coagulation, and contact systems [4,5]. The activity of these systems is directed to ensure the maintenance of basic homeostatic parameters in cells, tissues, and the whole organism. A steady increase in any kind of destruction in elderly people enhances the activity of these protective systems and is associated with an increase in permanent inflammatory states and immune dysfunction. This condition, which is known as inflammaging [6], represents a key risk factor for the increased appearance of life-threatening chronic diseases and adverse health outcomes [7].

During the inflammatory response, several host-derived cytotoxic agents are generated and released from activated immune cells and undergoing tissue cells [8]. These agents contribute to the destruction and elimination of foreign microorganisms and unwanted cells but can also disturb the integrity of healthy cells and tissues. Cytotoxic agents are usually well-controlled by corresponding antagonizing principles to avoid any damage to unperturbed cells in close proximity to inflammatory sites. Disturbances in the balance between cytotoxic agents and protective mechanisms favor chronic inflammatory states and disease progression [8]. Otherwise, the overexpression of antagonizing principles, as found, for example, in late-stage tumor cells, also markedly promotes the disease process [9].

In this review, the interplay between oxidant-based host-derived cytotoxic agents and protective mechanisms is analyzed in elderly persons. The question of how age-dependent alterations in energy metabolism are associated with the increased appearance of inflammatory states is addressed. Special attention is devoted to the enhanced production of reactive species from dysfunctional mitochondria, alterations in cellular redox homeostasis, and adaptations to hypoxia in older individuals. Finally, selected examples of how a disturbed balance between oxidant-based cytotoxic agents and antagonizing principles contributes to the pathogenesis of life-threatening diseases in older individuals are given.

## 2. Key Concepts About Inflammation in Older Individuals

### 2.1. The Concept of Inflammaging

The concept of inflammaging was developed from the network hypothesis of aging [10,11], which is based on the idea that, in elderly individuals, the ability of different anti-stress systems to counterbalance any destructive deviations in cells and organs is limited. A key point of inflammaging is that aging is associated with an increased activity of both the innate and activated immune responses in elderly persons [12]. Indeed, with advanced age, the serum levels of a variety of pro-inflammatory mediators increase, as shown for IL-6, IL-15, IL-8 [12,13,14], and coagulation factors like fibrinogen and von Willebrand factor [15,16]. Overall, this increase indicates the presence of a permanent inflammatory state that can be identified as low-grade chronic inflammation [17].

According to the inflammaging hypothesis, the balance between pro-inflammatory and anti-inflammatory agents is disturbed in older individuals [12]. This balance is crucial for resistance against any kind of disease. Important anti-inflammatory agents in elderly persons are TGF-β, cortisol, IL-10, and lipoxins [12,18,19].

Without going into detail, the concept of inflammaging summarizes key features characteristic of the chronic inflammatory process in older persons [7,12,17,20,21,22,23,24]. To obtain an answer for why inflammation persists under these conditions, it is necessary to analyze how molecular patterns and host-derived cytotoxic agents contribute to the development of long-lasting inflammatory states.

### 2.2. Molecular Patterns and Host-Derived Cytotoxic Agents in Inflammation

During inflammation, the activation of the immune system and its associated protective mechanisms is mandatory to eliminate pathogens and transformed and damaged cells and to induce repair processes. In humans, the inflammatory response is initiated by the presence of molecular patterns, which activate immune cascades via interaction with pattern recognition receptors (PRRs) [25]. These molecules result from pathogens (pathogen-associated molecular patterns (PAMPs)) and undergoing immune and non-immune cells of the host (damage-associated molecular patterns, DAMPs) [26,27]. Important DAMPs are heat-shock proteins, high-mobility group box 1 protein, fragments of hyaluronan, uric acid, heparan sulfate, free heme, and tumor DNA [28,29,30,31,32,33]. Acute inflammation is terminated when the number of molecular patterns falls below an unrecognizable level. In other words, inflammation becomes resolved when the pathogen load is considerably reduced and the damage of host cells is substantially limited.

Concerning major mediators and pathways, acute inflammation can be subdivided into two main phases, the initiation and propagation of inflammation and the resolution of inflammation. During the first phase of acute inflammation, as a result of the presence of molecular patterns, several immune cells are recruited to inflammatory sites and activated, whereby pro-inflammatory cytokines like IL-1, IL-6, IL-8, IL-15, and tumor-necrosis factor α (TNF-α) are involved in the attraction of neutrophils, monocytes, macrophages, T-cells, and others [34,35,36]. In addition, several acute-phase proteins are released into circulation, among which C-reactive protein (CRP) and serum amyloid A (SAA) serve as important biomarkers of the course of inflammation [34,37,38]. At inflamed loci, immune cells combat against pathogens and remove damaged cells and cell debris. In macrophages (here, subtype M1) and other immune cells, glycolysis is the preferred pathway for ATP production [39,40,41,42]. In the second phase of acute inflammation, immune cells are suppressed. There is now a dominance of repair processes and the synthesis of novel extracellular matrix components. These processes are driven by resolving mediators like transforming growth factor β (TGF-β), IL-10, vascular endothelial growth factor (VEGF), and lipoxins and are directed to restore the former tissue homeostasis [43,44,45,46,47,48,49]. The number of myeloid-derived suppressor cells (MDSCs) increases markedly [50,51]. The macrophage subtype M2, which actively promotes oxidative phosphorylation, dominates now [52,53]. The key events, mediators, and pathways of both main phases of acute inflammation are summarized in Table 1.

By which mechanisms the resolution of inflammation is initiated at inflammatory sites remains unknown. Under discussion is the enhanced formation of specific resolving mediators like TGF-β, IL-10, VEGF, and lipoxins, which drive accumulated immune cells into an anti-inflammatory and immunosuppressive state and promote tissue repair processes [43,44,45,46,47,48,49]. Another aspect concerns the prevailing dominance of apoptotic cell death processes over the uncontrolled release of cytotoxic agents from necrotic tissues and activated immune cells at inflammatory loci. In addition, the release of lactate from activated pro-inflammatory macrophages (M1 type), where glycolysis is the predominating pathway for the generation of energy equivalents, can induce a metabolic switch towards the anti-inflammatory M2 type in neighboring yet unperturbed invading macrophages and T cells [54,55,56]. As a result of this metabolic switch, lactate can be used as fuel for mitochondrial energy metabolism [54,55,56,57].

In order to better understand how an acute inflammatory response becomes persistent, it is necessary to analyze the roles of different host-derived cytotoxic agents in these processes [8]. Molecular patterns are able to initiate inflammatory events. Recruited and activated immune cells are directed to inactivate and kill pathogens and transformed cells and to destroy and remove damaged cell material. In these activities, immune cells apply different cytotoxic agents, which can also damage yet unperturbed cells at inflammatory sites. Healthy cells and tissues are equipped with numerous antagonizing principles, which inactivate immediately cytotoxic agents and, thus, limit their unwanted destructive actions. The cytotoxic products of activated neutrophils and their antagonizing principles are listed in Table 2. Other major sources of cytotoxic agents are other immune cells, defective mitochondria, blood hemorrhages, and damaged muscles. In addition to the cytotoxic agents listed in Table 2, other important cytotoxic agents are peroxynitrite, free heme, lipid hydroperoxides, mast cell proteases, angiotensin, bradykinin, and matrix metalloproteases [8].

The interplay between cytotoxic agents and antagonizing principles greatly determines the further fate of inflammation [8]. Inflammation becomes resolved when antagonizing principles properly control the action of cytotoxic agents and deactivate them. Any disturbance in the balance between cytotoxic agents and antagonizing principles can result in chronic inflammatory states due to the inability of protective mechanisms to efficiently eliminate or inactivate host-derived cytotoxic agents [8]. As a result of this inability, novel DAMPs and antigens are released by the action of cytotoxic agents and the inflammatory process continues. A scheme of the interplay between molecular patterns and cytotoxic agents in persistent chronic inflammatory states is given in Figure 1.

In older people, low-grade chronic inflammation dominates. It is evident that the terms pro-inflammatory and anti-inflammatory agents used in the concept of inflammaging [12] correspond to the predominating mediators in the two main phases of acute inflammation. During the initiation and propagation of inflammation, pro-inflammatory agents play an important role, and anti-inflammatory mediators are typical of the resolution of inflammation. In analyzing inflammatory processes in older individuals, the main focus has been directed toward the behavior of different pro- and anti-inflammatory markers in association with selected disease processes. Considering the imbalance between cytotoxic agents and antagonizing principles in the development of different age-related disease scenarios, in this review, the attention is focused on the role of reactive species, the loss of control over the sequestration of transition metal ions, and the diminution and exhaustion of haptoglobin and hemopexin. Oxidant-related cytotoxic agents have a great potential to disturb the metabolic processes within cells during aging and, thus, to contribute to development of associated disease scenarios in older people.

In analyzing the molecular mechanisms of age-related metabolic alterations and conditions for the development of diseases, changes in energy metabolism provide a fundamental clue to better understand the disturbed interplay between oxidant-based cytotoxic agents and antagonizing principles in older individuals.

## 3. Peculiarities of Energy Metabolism and Oxidative Stress in Older Individuals

### 3.1. Energy Metabolism Under Normoxic Conditions

In humans and higher animals, adenosine triphosphate (ATP) is the master molecule providing energy equivalents for numerous energy-intensive processes, such as muscle contraction, kinase activities, active transport, intra- and extracellular signaling, and others [87,88,89]. In human cells, ATP can be generated both in the cytosol by anerobic glycolysis and most of all within the mitochondria by oxidative phosphorylation. The consumption of one mole of glucose during dioxygen-independent cytosolic glycolysis yields two moles of ATP. In mitochondria, the production of ATP is linked to the tricarboxylic acid (TCA) cycle (Krebs cycle) and the electron transport chain through mitochondrial complexes I–IV. During these processes, a pH gradient and an electrical potential are generated across the inner mitochondrial membrane, which are essential for the proper functionality of ATP synthase. These metabolic pathways highly depend on the presence of dioxygen and yield about 30–36 moles of ATP per mole of initial substrate [90].

Several pathways can supply substrates that enter the TCA cycle for subsequent ATP production. The degradation of glucose by glycolysis yields pyruvate and further acetyl-CoA, which enter the TCA cycle. The latter substance is also produced during the β-oxidation of long-chain fatty acids. Which type of fuel molecule is utilized for ATP production in cells depends on cell function and physiological context [91]. In cardiac muscle cells and adipose tissue, the β-oxidation of fatty acids predominates to produce ATP. Otherwise, glucose is the preferred fuel molecule in cells of the central nervous system. The degradation of free amino acids contributes to about 10–15% of ATP generated via oxidative phosphorylation [92]. After deamination, the remaining carbon skeleton of amino acids is converted into intermediates, which can enter the TCA cycle at several places.

Although mitochondrial oxidative phosphorylation is more efficient than cytosolic glycolysis concerning the amount of ATP generated per mole of initial substrate, glycolysis predominates as the main process of energy production in some immune cells (e.g., in activated neutrophils [42], activated macrophages (M1 type) [40], maturating dendritic cells [41], T_h_1, T_h_2, and T_h_17 lymphocytes [39,93,94], muscle fibers type IIB [95], and under certain (patho)physiological conditions (e.g., starvation and prolonged exercise)). ATP formation by glycolysis is more rapid than that within mitochondria. Moreover, glycolysis can be further intensified by the up-regulation of the enzymes involved and the fast supply of additional glucose from glycogen stores [96].

Starvation, prolonged intense exercise, some diets, dioxygen deprivation, and disease conditions can deplete glycogen stores in the liver and muscles [96,97]. As a result, ketone bodies are formed and the de novo synthesis of glucose is accelerated in the liver. Both mechanisms contribute to ATP production. Ketone bodies (acetoacetic acid, acetone, and β-hydroxybutyrate) are derived from fatty acids in the liver and are converted after their transport to tissues into acetyl-CoA, which enters the TCA cycle [98]. In the liver and, to some extent, also in the kidneys, glucose can be generated by gluconeogenesis, whereby lactate, glycerol, glutamine, and alanine are the starting sources for this pathway [92,99]. By these mechanisms, ATP maintenance is ensured to some extent, even in the short term, and mild disturbances in energy supply and different (patho)physiological conditions are bypassed. Major pathways of cellular energy metabolism are schematically depicted in Figure 2.

### 3.2. Deviations in Energy Metabolism in Older Individuals

In older individuals, the production of energy equivalents by oxidative phosphorylation can be highly disturbed by dioxygen deficiency, the presence of dysfunctional mitochondria, and a lower number of these organelles [100,101,102]. Under these conditions, the TCA cycle and mitochondrial electron transport chain are unable to generate high amounts of ATP. During aging, it may occur that glycolysis is highly up-regulated, with lactate as a final product. Deficient mitochondrial processes can lead to the enhanced formation of superoxide anion radicals (O_2_^•−^) and other reactive species like hydrogen peroxide (H_2_O_2_), Fe^2+^, hydroxyl radicals (HO^•^), and peroxynitrite (ONOO^−^), which all are derived from reactions of superoxide anion radicals. Important energy metabolism alterations in older individuals are also highlighted in Figure 2.

Several aspects are under discussion that contribute to the reduced capacity of mitochondria for ATP production in older individuals. (i) Mitochondrial DNA is more prone to damage than nuclear DNA [103]. The lack of histones within mitochondria and a less efficient nucleic acid repair machinery are responsible for the higher mutation rate of mitochondrial DNA [104]. (ii) An enhanced production of mitochondria-derived reactive species is another key mediator disturbing the enzymes and proteins involved in oxidative phosphorylation. Hypoxic conditions are a major contributor to the increased generation of mitochondria-derived reactive species. (iii) In skeletal muscle, the number of mitochondria declines gradually with an increasing age [105]. (iv) The rate of cell apoptosis with defective mitochondria also increases with an increasing age [106].

Age-related alterations in mitochondria are also reflected by the decline in NAD^+^ [107,108] and α-ketoglutarate [109,110]. NAD^+^ is involved in many enzymatic reactions within mitochondria. This decline impairs, for example, the activity of sirtuins, which play an important role in the maintenance of mitochondrial homeostasis [111,112,113,114]. A decreased level of α-ketoglutarate affects the epigenetic landscape in several tissues [110,115,116].

Regarding the molecular mechanisms of energy metabolism in mitochondria, accelerated aging is mainly favored by the activation of the mTOR pathway and the suppression of autophagy by an increased formation of aspartate from oxalacetate or an increased yield of pyruvate, which is further converted to acetate [117]. These mechanisms are supported by high-caloric food and a sedentary lifestyle. Further factors contributing to accelerated aging are the formation of advanced glycation end products due to an enhanced glycolysis rate, impaired repair of DNA hypermethylation induced by increased succinate levels, and reduced maintenance of the stem cell pool by an increased level of α-ketoglutarate [117]. Several alterations in the activity of TCA cycle enzymes are responsible for the aforementioned processes. Oxidative stress is associated with decreased activities of succinate dehydrogenase, α-ketoglutarate dehydrogenase, and aconitase and favors the conversion of oxalacetate and glutamate by aspartate aminotransferase into α-ketoglutarate and aspartate. Hence, enhanced levels of aspartate, succinate, and α-ketoglutarate occur as a result [117]. Aconitase contains a 4Fe–4S cluster at active sites. Under stress conditions, superoxide anion radicals are known to release Fe^2+^ from this iron–sulfur cluster [118,119].

A decreased energy expenditure by oxidative phosphorylation can be compensated by increased anaerobic glycolysis to yield pyruvate, which is further converted by lactate dehydrogenase (LDH) into lactate. This reaction sequence is activated under hypoxic conditions and in cases of high energy demand [120,121]. As the LDH-driven conversion of pyruvate into lactate is accompanied by the reduction of NADH, the resulting NAD^+^ stabilizes the cytoplasmic redox status [122,123]. Lactate is secreted from cells together with H^+^. Once released, lactate is disposed of by other cells within the tissue or transported via the bloodstream mainly to the liver, brain, heart, and operating skeletal muscle, where it is taken up and serves as fuel [124,125,126] or is used for gluconeogenesis in the liver [127]. Furthermore, lactate promotes the release of Mg^2+^ from the endoplasmic reticulum, favors gene expression by interaction with histones, and inhibits lipolysis via ligand binding to the G-protein GPR81 [128,129].

Lactate is an anti-inflammatory agent. At sites with enhanced lactate levels, monocytes and macrophages are reprogramed into an anti-inflammatory M2 type [56,57]. In human monocytes and macrophages, extracellular lactate shifts the energy metabolism from glycolysis to oxidative phosphorylation. While short-term lactate exposure has limited effects on cytokine production, long-term exposure reprograms immune cells to anti-inflammatory responses [57]. Lactate inhibits monocyte migration and cytokine release [130,131].

When glycolysis prevails for ATP production, mitochondria serve mainly as biosynthetic organelles to yield initial components for the synthesis of lipids, nucleotides, and non-essential amino acids [132]. The uptake of glutamine increases markedly under these conditions. The stepwise conversion of glutamine into α-ketoglutarate provides the basis for the synthesis of biological molecules independent of glucose metabolism. For example, α-ketoglutarate is converted via the intermediate formation of citrate into acetyl-Co-A, which is necessary for lipogenesis [133].

The aforementioned deviations in energy metabolism in older individuals affect general physiological processes, induce numerous adaptations to altered cellular metabolic routes, and promote inflammatory reactions by the release of DAMPs from defective cells. In particular, oxidative stress and oxidants, as cytotoxic agents, play an increasing role under these conditions.

### 3.3. Responses to Hypoxia and Oxidative Stress

In order to better understand the real nature of inflammatory processes in individuals of advanced age, it is necessary to focus on the basic mechanisms of how cell and tissue homeostasis can be maintained over a long time under conditions of increased impacts. In elderly persons, these impacts result predominantly from deficits in dioxygen and nutrient supply. Hypoxia is a key factor affecting metabolic routes in elderly persons [134,135,136]. Without considering external reasons for generalized hypoxia such as dioxygen deficiency in breathing gas (e.g., people at high altitudes), at an advanced age, hypoxic conditions are often the consequence of a diminished blood flow, lung problems, or a reduced capacity of blood cells to carry dioxygen. In addition, health problems and enhanced blood pressure values are other major players in the formation of hypoxic conditions.

In the affected tissue regions, hypoxia causes the activation and stabilization of cytosolic hypoxia-inducible factor 1α (HIF-1α), which up-regulates glycolysis, down-regulates mitochondrial oxidative phosphorylation, and favors the enhanced formation of reactive species in dysfunctional mitochondria [137,138,139]. In highly vascularized tissues, HIF-1α is replaced by HIF-2α during prolonged hypoxia [140,141].

HIF-1α is a master regulator of glycolysis in muscles and many other cells. Cytosolic HIF-1α is usually tagged by prolyl hydroxylase for proteasome degradation [142,143]. Essential cofactors for active prolyl hydroxylase are O_2_, Fe^2+^, and α-ketoglutarate [144]. This proteasomal degradation of HIF-1α is lost under hypoxic conditions due to a lack of O_2_ [137]. In addition, HIF-1α is known to up-regulate the transcription of several enzymes promoting glycolysis, such as hexokinase II (HK2), pyruvate kinase M2 (PKM2), lactate dehydrogenase A (LDHA), and pyruvate dehydrogenase kinase-1 (PDHK1). The latter enzyme is a negative regulator of pyruvate dehydrogenase (PDH) [138]. As a result, pyruvate is mainly metabolized to lactate, but not to acetyl-CoA in hypoxic cells.

In addition to the stabilization of cellular redox processes, long-lasting adaptations to hypoxia mainly concern the overexpression of proteins antagonizing oxidative stress. The activation of the stress-sensitive transcription factor Nrf2 protects cells against damaging reactions of reactive species and related oxygen-based cytotoxic agents [145,146,147]. Under stress conditions, superoxide dismutase, catalase, glutathione peroxidase, and other protective antioxidative proteins can be up-regulated. This at least partially or more efficiently counter-regulates stress-related oxidant-based cytotoxic agents. Several potential scenarios determine the further fates of affected cells and tissues. First, cell metabolism can be stabilized by the up-regulation of protective antioxidant systems, meaning that cells survive, as found in many cancers. Second, the processes of autophagy and mitophagy are enhanced to remove damaged cell material and defective organelles. Third, last but not least, cells are driven into apoptosis or necrosis. These scenarios have fundamental consequences for the (patho)physiological features of hypoxic areas.

Enhanced expressions of hypoxia-induced factors (HIFs) cause numerous gene-induced adaptations to stabilize the cellular redox status, to resist against oxidative stress, and to reprogram cell metabolism. Enhanced intracellular values of reactive species and other oxidant-based cytotoxic agents activate the transcription factor nuclear factor erythroid 2-related factor 2 (Nrf2), which plays a key role in the enhanced synthesis of antioxidant proteins [145,146]. Under normal physiological conditions, Nrf2 is complexed with the by Kelch like-ECH-associated protein 1 (KEAP1) and Cullin 3, which cause the ubiquitination and degradation of Nrf2 [147]. Cellular stress releases Nrf2 from this complex and promotes nuclear transcriptional processes by Nrf2. Genes activated by Nrf2 code proteins, which are involved in carbohydrate metabolism, including NADPH generation, redox regulation, and antioxidant defense systems like glutathione- and thioredoxin-mediated reactions, heme detoxification, iron metabolism, lipid metabolism, proteasomal degradation, autophagy, apoptosis, and biotransformation and detoxification [148].

### 3.4. Redox Regulation and Antioxidative Defense in Older Individuals

Many physiological cell functions depend on the presence of a reducing milieu. A high cytoplasmic level of NADPH is mandatory to ensure redox homeostasis in cells. NADPH is the major electron source within cells [123]. It is oxidized to NADH^+^ by glutathione reductase [149] and thioredoxin reductase [150,151], keeping both glutathione (GSH) and thioredoxin (Trx) in their reduced states. In quiescent cells, the ratio of [NADPH]/[NADP^+^] is approximately 100:1 [123]. Two enzymes of the pentose phosphate pathway, which runs parallel to glycolysis, namely glucose-6-phosphate dehydrogenase and 6-phosphogluconate dehydrogenase, reduce NADP^+^ to NADPH [152]. An overview of the basic mechanisms of cellular redox homeostasis and the potential alterations occurring under stress situations is given in Figure 3. In contrast, in the redox couple NAD^+^/NADH, the oxidized form predominates, giving values between 1:10 and 1:1000 for the ratio of [NADH]/[NAD^+^] [123]. Interconversions between members of both nicotinamide-based redox couples are controlled by NADH kinases and NADPH phosphatases [153]. By these reactions and the presence of numerous cytosolic binding sites for NADH and NADP^+^ [123], both redox couples are kept far from equilibrium. These conditions allow them to fulfil key physiological functions in the maintenance of redox homeostasis.

With an advanced age, the increasing significance of glycolysis has several consequences for redox status. In addition, cytoplasmic and mitochondrial stress increase too. The higher demand for NADPH can be compensated for by the enhancement of the pentose phosphate pathway. The enhanced formation of lactate from pyruvate is coupled with the reduction of NADH to NAD^+^. The latter species is a cofactor for numerous enzymes, including sirtuins, which are important modulators of mitochondrial homeostasis and aging [111,112,113,114]. Both the reduced formation of novel NAD^+^ and higher activity of NAD^+^-consuming enzymes contribute to an age-dependent decline in NAD^+^ [102,107,108]. A reduced level of NAD^+^ is associated with defects in oxidative phosphorylation [154,155,156,157,158].

The glutathione system is involved in the removal of lipid hydroperoxides, peroxynitrite, and hydrogen peroxide by glutathione-peroxidase-mediated reactions, the reduction of disulfides, and the reduction of glutaredoxin [159]. Glutathione reductase converts oxidized glutathione (GS-SG) to its reduced form (GSH). In addition, GSH can be supplied by de novo synthetic processes. In elderly individuals, the disturbance of components of the redox metabolism highly depends on the intactness of immune functions. Thus, in this cohort, improved immune functions correlate with higher values of glutathione peroxidase, glutathione reductase, GSH, and the GSH/GS-SG ratio [160]. Impaired immune functions are associated with lower levels of these parameters. An investigation of large groups of healthy humans revealed a gradual decline in GSH content with an increasing age in blood and blood cells [161,162,163]. A gradual decrease in GPx activity has been reported in female persons after age 65 [164]. In the brain of older persons, impaired glutathione is associated with different neurological disorders [165,166].

The thioredoxin system is another major contributor to redox homeostasis in cells. Reduced thioredoxin (Trx) is not only an essential factor to reduce cellular oxidized proteins such as oxidized peroxiredoxins, ribonucleotide 5′-diphosphatase, and methionine sulfoxide [159], but it also plays an important regulatory role as a cofactor for several enzymes and transcription factors. In other words, Trx is regarded as a biomarker for age-related diseases, oxidative stress, inflammation, and cellular senescence [167]. For example, reduced Trx forms complexes with apoptosis-signal-regulating protein 1 (ASK1) and phosphates and tensin homolog (PTEN), keeping these binding partners in an inactive state [168].

Enhanced cellular damage, redox imbalance, and disease progression are linked to diminished Trx and the activation of the thioredoxin-interacting protein (Txnip) [169] and proteins like ASK1 and PTEN, which are both involved in apoptosis induction. In cells, Txnip forms a complex with reduced Trx via the formation of a disulfide bond [170]. Oxidative stress and Trx deficiency lead to the decomposition of this inhibitory complex and allow Txnip to interact with other target molecules. Txnip activates the (NOD)-like receptor protein-3 (NRLP3) inflammasome complex via interaction with the proapoptotic protein ASK1 [171,172]. The inflammasome complex contributes to mitochondrial-stress-induced apoptosis via the expression of inflammatory cytokines. In addition, Txnip binds to phosphatase and tensin homolog (PTEN) and, thus, favors apoptosis induction through the regeneration of PIP_2_ [168]. Txnip also suppresses glycolysis and promotes oxidative phosphorylation. Glucose transporters 1 and 4 are up-regulated by Txnip [173,174,175]. The expression of Txnip in blood precursor cells implies a role of Txnip during hematopoiesis [176].

Txnip is up-regulated during aging in primary human cells [177]. This up-regulation is associated with increased oxidative stress, DNA damage, and a reduced lifespan [177]. In diabetes patients, Txnip is up-regulated too [178,179,180]. Txnip is also implicated in the pathogenesis of Alzheimer’s disease and Morbus Parkinson [181,182,183]. In brain samples from patients with Alzheimer’s disease, less neuronal Txnip was observed in comparison to unperturbed brains. However, an increased number of microglia cells was found in these samples, which were associated with Txnip-positive plaques [184].

Otherwise, Txnip is a negative regulator of cell proliferation. Up-regulated Txnip inhibits cyclin A and several other factors promoting the cell cycle [169]. Cancer cell growth is associated with decreased Txnip, increased glycolytic flux, and the enhancement of proliferative processes [185,186].

In addition to the aforementioned components of the glutathione and thioredoxin systems, further proteins and enzymes contribute to defense against oxidative stress. In drosophila and some other invertebrate models, the over-expression of superoxide dismutase (SOD) has been associated with protection against reactive species and an extended lifespan [187,188]. Otherwise, SOD activity is dispensable for normal animal aging but is useful in stress situations [189]. In humans, the potential relationship between SOD and aging remains unclear. In older women, a higher level of plasma SOD activity was linked to a lower overall mortality, but this was not the case in older men [190]. SODs are essential to catalyze the dismutation of superoxide anion radicals, which mainly result from dysfunctional mitochondria, activated immune cells, and other sources, into dioxygen and hydrogen peroxide. In this way, SODs prevent the side reactions of superoxide anion radicals with nitrogen oxide to yield the powerful oxidant peroxynitrite [191,192] and with iron–sulfur clusters of some mitochondrial proteins, resulting in the release of catalytically active Fe^2+^ [118,119].

Hydrogen peroxide is inactivated by glutathione peroxidases, catalase, and peroxiredoxins. Catalase malfunction or deficiency is hypothesized to play a role in the pathogenesis of age-related diseases such as diabetes mellitus, hypertension, neurodegenerative disorders, vitiligo, and some others [193]. Age-related alterations in the functions of peroxiredoxins have mainly been obtained from experiments with mice [194]. Peroxiredoxin 3 (Prx3), the mitochondrial form of peroxiredoxins, is involved in the maintenance of mitochondrial redox homeostasis [195,196]. As the expression of Prx3 declines with age, mitochondrial functions and homeostasis are impaired [197]. Prx1 contributes to dampening telomere shortening, a key age-dependent process [198,199,200].

Iron is an essential cofactor of many proteins. Within mitochondria, iron is part of many proteins in the form of iron–sulfur clusters and is a central ion in the heme group of heme proteins. As free iron ions have a great potential to catalyze damaging reactions in cells, all aspects of iron transport, utilization, and metabolism are usually well-controlled [201,202]. In cells, transferrin receptor (iron uptake), ferritin (iron storage), and ferroportin (iron export) are major players to ensure iron homeostasis and utilization [66,67,68,69]. In older individuals, an increased ferritin level is associated with the presence of a chronic inflammatory state [203]. Otherwise, lower ferritin levels are found in patients with an iron deficiency [203].

Similar conclusions concerning transport, utilization, and metabolism can be drawn for copper ions. In blood, copper is mostly bound and transported by ceruloplasmin, followed by serum albumin [204,205]. Copper-containing ceruloplasmin detoxifies Fe^2+^ by oxidation to Fe^3+^ and, thus, contributes to the maintenance of iron homeostasis [206]. A glycosylphosphatidylinositol (GPI)-anchored form of ceruloplasmin [207] is present in astrocytes, oligodendrocytes, hepatocytes, macrophages, and epithelial cells in the pancreas and retina [208,209] and is part of the iron efflux machinery [210]. GPI-anchored ceruloplasmin oxidizes intracellular Fe^2+^ to Fe^3+^, which is excreted by ferroportin and further safely transferred to transferrin [211]. Patients deficient in ceruloplasmin develop massive iron depositions in the brain, liver, and other organs and exhibit neurological symptoms and motor deficits [212,213,214]. Similarly, ceruloplasmin-knockout mice are characterized by iron accumulation in astrocytes, ferritin expression, and neurodegeneration in the central nervous system [207,215].

An uncontrolled increase in the free labile pool of iron ions can lead to ferroptosis. In this kind of cell death, enhanced levels of free iron ions catalyze the oxidation of lipid hydroperoxides, which has fatal consequences on the barrier function of biological membranes. These mechanisms are promoted by glutathione deficiency and a decreased activity of glutathione peroxidase 4 [216,217]. The latter enzyme is, together with its cofactor glutathione, responsible for the detoxification of lipid hydroperoxides [62]. Another player in the removal of lipid hydroperoxides and the prevention of ferroptosis is peroxiredoxin 6 [218,219].

In plasma, haptoglobin binds hemoglobin after its release from red blood cells and myoglobin resulting from damaged muscles [220,221]. The haptoglobin–heme protein complex is removed from circulation by spleen and liver macrophages [222,223]. Both released hemoglobin and myoglobin can liberate the powerful oxidant ferriprotoporphyrin IX, also known as free heme [224]. Hemopexin inactivates free heme by forming a high-affinity complex with free heme and clearing this complex in the liver [225]. In elderly humans, the blood level of haptoglobin is elevated [226]. Another study revealed enhanced serum values for the haptoglobin–hemoglobin complex with age [227]. These data indicate an enhanced leakage of red blood cells and maybe problems with the clearance of the haptoglobin–heme protein complex in older individuals.

In the cell cytoplasm, hemoxygenase 1 (HO-1) degrades free heme to biliverdin, carbon monoxide, and Fe^2+^. The role of HO-1-derived Fe^2+^ is controversially discussed [228,229]. Low amounts of Fe^2+^ can be stored in ferritin and induce the synthesis of additional ferritin [230]. Higher levels of Fe^2+^ and an overload of ferritin with iron can promote ferroptosis [231]. A decline in HO-1 expression occurs with an increasing age in the macular and peripheral retinal pigment epithelium [229,232].

### 3.5. Activation of Proteolytic Systems in Older Individuals

Normally, the processes of protein synthesis and degradation are in equilibrium in living tissues. With an increasing age, however, degradative mechanisms prevail. This is mainly caused by disturbances in energy metabolism, dioxygen and nutrient deficiencies, and increased defects in cellular constituents and organelles in association with stress situations.

There are several levels to how cell organelles and cells respond to stress situations. The mitochondrial unfolded protein response (UPR) is activated when the number of unfolded proteins is increased beyond a certain threshold. In this pathway, genes are activated that favor the translation of proteins involved in the detoxification of reactive species, supporting the correct folding and removing misfolded proteins [102]. In mouse models, positive effects of enhanced levels of NAD^+^ have been demonstrated on improvements in the UPR and mitochondrial homeostasis [154,233]. As the UPR links mitochondrial and nuclear genomes, the dysregulation of this pathway has been reported in several age-related human diseases like sarcopenia and Morbus Alzheimer [234,235,236,237].

In stress situations, the processes of mitochondrial membrane dynamics, known as mitochondrial fusion and fission, allow for diluting and segregating damaged organelles in order to ensure homeostasis and survive [238,239]. The dynamin-related protein 1 plays an important role in fission regulation [102]. In mouse models, reduced activity of this protein is impaired with an increasing age [240,241].

In cells, defective proteins are recognized and degraded by the ubiquitin–proteasome system [242,243]. With an increasing age, misfolded and damaged proteins accumulate [244]. During aging, the induction of chaperones is impaired [245] and proteasomal activity declines, as evidenced in different cells and tissues [246,247,248]. For example, age-related decreases in the activities of the 20S and 26S proteasome complexes have been reported [249,250,251].

Damaged cell organelles, dysfunctional cytoplasmic and structural components, protein aggregates, and sometimes invading pathogens are preferentially cleared by autophagic mechanisms by the means of lysosomes [252,253]. The selective removal of damaged mitochondria by autophagy is known as mitophagy. Both ubiquitin-dependent and -independent pathways of mitophagy are differentiated [102]. In mice and humans, a decline in mitophagy has been observed in several tissues upon aging [254,255,256,257,258].

Defective mitochondria are linked to immune activation via the release of DAMPs. Putative mitochondrial DAMPs are mitochondrial DNA, cardiolipin, and N-formyl peptides resulting from the translation of mitochondrial-encoded proteins [259]. A gradual increase in mitochondrial DNA occurs in the blood of elderly persons [260]. Mitochondria-derived DAMPs are involved in NLRP3 inflammasome activation and, consequently, in the release of pro-inflammatory cytokines like IL-1β and IL-18 and apoptosis induction [261,262,263].

Both the ubiquitin–proteasome system and autophagy are intensified in cachectic patients [264]. Cachexia is associated with progressive skeletal muscle wasting and substantial weight loss as a consequence of underlying illness and is an important comorbidity in cancer patients [265,266,267], as well as in individuals of an advanced age [268,269,270].

### 3.6. Inflammaging and Necrotic Cell Death

Inflammaging is characterized by an increasing number of factors secreted from senescent cells. These factors are summarized as a senescence-associated secretory cell phenotype. They include pro-inflammatory cytokines, chemokines, growth factors, proteases, extracellular matrix components, and some others [24]. The pathophysiological consequences of inflammaging are very complex. On the cellular level, increased oxidative stress is assumed to be a crucial factor for the accumulation of numerous structural and functional defects and the initiation of novel inflammatory events [271,272,273]. This interplay between oxidative stress and inflammation during aging is also associated with increased dysfunctional cell death. In addition to apoptotic cell death, which usually occurs without further disturbances to neighboring cells and tissues, different forms of necrosis increase with age. These forms include ferroptosis, necroptosis, pyroptosis, and others [270]. During necrotic cell death, both cytotoxic agents and different DAMPs are released. These agents contribute to further tissue damage and trigger novel inflammatory events.

A few examples underline the pathophysiological significance of necrotic cell death in older persons. In aged livers, an increase in necroptosis, a regulated form of programmed necrosis, contributes to chronic liver inflammation and the development of liver cirrhosis [274]. The risk for pyroptosis increases in aged cells, as these cells are more susceptible to external stimuli that can cause cell damage [275]. Cytokine release and inflammasome activation by pyroptosis have been linked to the development of atherosclerosis and neurodegenerative diseases [276,277]. Iron-dependent lipid peroxidation leads to ferroptotic cell death. This mechanism has been discussed to contribute to the pathogeneses of neurodegenerative diseases, autoimmune disorders, and cardiovascular diseases [278,279].

Increasing immunosuppression is another key feature of inflammaging. During the resolution of inflammation, transient immunosuppression allows for down-regulating all inflammatory processes, including the apoptosis of immune cells, to induce the de novo formation of matrix and tissue components and restore normal tissue homeostasis [43,44]. Under chronic conditions, as shown in inflammaging, immunosuppressive states can be pronounced more strongly and take longer. Both enhanced expressions of host-derived cytotoxic agents and decline, exhaustion, or inactivation of the corresponding antagonizing principles favor the chronic inflammatory process [8]. Under these conditions, different comorbidities develop in persons of an advanced age. Moreover, long-lasting immunosuppression dampens general immune functions and favors infections with commensal and mutualistic pathogens [280,281,282,283,284]. In the worst case, organ failure, sepsis, and septic shock develop, as may occur in immunocompromised individuals [285,286].

## 4. Age-Related Alterations in Protection Against Oxidant-Based Cytotoxic Agents in Selected Disease Scenarios

### 4.1. Diseases of the Cardiovascular System

Disturbances in blood vessels and heart function are very common in elderly persons. Most of all, the physiological functions in older individuals are affected by increased blood pressure, hypoxia, alterations in blood vessel elasticity, processes of intravascular hemolysis, internal hemorrhages, and inflammatory processes in vessel walls. Common health problems are atherosclerotic vascular diseases, including coronary artery disease, myocardial infarction, cerebrovascular disease, and stroke. Further serious blood flow disturbances are caused by tissue damage due to ischemic and reperfusion events, thrombosis, reduced wound healing, and vasculitis.

The accumulation of cholesterol-rich low-density lipoproteins in a damaged endothelium is a hallmark of atherosclerosis. Lipid oxidation, plaque formation, the attraction of immune cells, alterations in endothelial cells, and the development of foam cells are further characteristics of this disease [287,288,289,290,291,292]. As cholesterol crystals and numerous DAMPs are present in such plaques, novel inflammatory cascades are continuously activated, contributing to the chronic inflammatory process.

Increasing age is considered to be a risk factor for the development of atherosclerotic vascular diseases [293,294,295,296]. The molecular details of the pathogenesis of atherosclerosis are not fully understood. Nevertheless, it is evident that oxidative processes play an important role in disease progression. In addition to cholesterol and lipid oxidation, the involvement of myeloperoxidase, a heme protein released from recruited neutrophils, is discussed in plaque formation. This enzyme is found in atherosclerotic plaques [297,298], produces the powerful oxidant hypochlorous acid, and catalyzes numerous other oxidative processes [299]. The potential participation of free heme presents another route discussed in atherosclerotic plaque formation [300]. As free heme is known to easily penetrate hydrophobic areas of proteins, membranes, and lipoproteins, it contributes to endothelial damage and lipoprotein oxidation during atherosclerosis [300,301].

The production of nitric oxide (NO) by NO synthases in the vessel wall is crucial for vessel relaxation and the regulation of blood flow. A decreased bioavailability of NO is observed by enhanced processes of intravascular hemolysis from red blood cells or rhabdomyolysis from damaged muscles [221,302]. Both released hemoglobin and myoglobin are known to be rapidly oxidized by NO [303,304,305]. In addition, myeloperoxidase, which can be attached to the inflamed endothelium, interacts with NO and, thus, diminishes the bioavailability of NO [306].

Another complication in cardiovascular diseases is the formation of internal hemorrhages. This bleeding is associated with uncontrolled hypertension, problems in blood clotting, traumata, aneurysms, a loss of blood vessel elasticity, reduced arterial compliance with an increasing age, and weakening of collagen linkages to the vessel wall [307,308,309]. In newly formed hemorrhages, there is a loss of glucose, dioxygen, and other nutrients with time. Concomitantly, the intravascular hemolysis of red blood cells increases and protective haptoglobin and hemopexin are exhausted. The resulting cytotoxic free heme can induce numerous damaging reactions at adjacent tissue areas [301,310,311].

Thrombotic complications can result from the rupture and erosion of atherosclerotic plaques and from deep vein thrombotic events. With increasing age, the shifted balance between coagulation proteins and anticoagulant factors increases the risk for the development of venous thrombosis [312,313,314].

### 4.2. Diabetes Mellitus

The majority of people over 65 years have prediabetes or diabetes, whereby diabetes type 2 predominates [315]. Diabetes is associated with many serious disease complications like vascular problems, neuropathy, nephropathy, and retinopathy [316]. Hyperglycemia leads to dysfunctions in red blood cells, endothelial cells, and other cells. These cells are known to accumulate glucose in an uncontrolled fashion, independent of the presence of insulin. As a result, the glycation of proteins in these cells increases beyond the level found in healthy individuals and different advanced glycation end products accumulate [317,318].

In red blood cells, the glycation of hemoglobin, as measured by the HbA1c value, serves as a long-term marker for hyperglycemia [319,320]. In these cells, part of glucose is converted into sorbitol, which cannot leave the cells [321]. Accumulated sorbitol contributes to an increased stiffness and diminished deformability of red blood cells [322,323,324]. Hence, these cells become osmotically instable and intravascular hemolysis increases [323]. Vascular endothelial cells develop an inflammatory phenotype upon hyperglycemia [325,326]. These alterations diminish blood flow by a reduced bioavailability of nitric oxide, decreased vasorelaxation, impaired fibrinolytic activities, changes in the cellular cytoskeleton, and the thickening of the basal lamina [326,327,328]. In addition, the release of the von Willebrand factor from endothelial cells exposed to high levels of glucose can initiate thrombolytic events and adverse cardiovascular complications [329,330,331].

### 4.3. Cancer

With advanced age, the incidence of the development of different kinds of cancers increases. Which concrete conditions contribute to cancer development in a given patient remains unknown. Key factors for cancer development in older individuals are increasing hypoxia in tissues and underlying problems in immune defense due to the presence of low-stage chronic inflammation.

In patients with advanced cancers, the host’s own protective mechanisms are triggered in a way that can support cancer growth and metastasis formation rather than the recognition and elimination of cancer cells by the immune system [332,333]. In cancer cells, antioxidative proteins and mechanisms stabilizing cellular redox homeostasis are highly up-regulated [9]. With these adaptations, cancer cells tolerate enhanced values of reactive species. This allows them to survive even therapeutic approaches like radio- and chemotherapy. Moreover, in the tumor microenvironment (TME), invading immune cells are in a permanent state of immunosuppression [334,335,336,337], which corresponds to down-regulated immune functions during the resolution phase of acute inflammation [9]. Cancer cells release several factors such as tumor-necrosis factor β, IL-10, vascular endothelial growth factor, and components contained in secreted exosomes, which promote immunosuppression and support the resolution of inflammation [338].

A high up-regulation of glycolysis with lactate as a final product is a metabolic hallmark in cancer cells. Anaerobic glycolysis supplies the necessary energy equivalents in these cells instead of dioxygen-dependent mitochondrial oxidative phosphorylation. Secreted lactate can be used in the TME as fuel by invading macrophages to trigger them into assuming the anti-inflammatory subtype M2 [56,339]. Enhanced expressions of carbonic anhydrases on the external surfaces of cancer cells and the up-regulation of several transport proteins in the plasma membrane acidify the pH value in the TME and decrease the cytoplasmic proton concentration [340,341,342,343]. The resulting mild alkalization in cancer cell cytosol is typical of proliferating cells. In other words, the enhancement of glycolysis and subsequent metabolic alterations support cancer growth, metastasis, and invasion [344,345].

### 4.4. Neurodegenerative Diseases

Neuronal plaque deposits determine the morphological picture of neurodegenerative diseases like Alzheimer’s disease, Parkinson’s disease, and others. These diseases often appear in people of an advanced age. The accumulation of fibrillated proteins coincides with a progressive loss of mental abilities due to the loss of functional neurons and synaptic linkages.

In Alzheimer’s disease, large extracellular deposits of amyloid β and hyperphosphorylated τ protein accumulate [346,347]. However, the molecular details supporting fibrillation are largely unknown. Disturbances in the endosomal–lysosomal and autophagic pathways contribute to the processing of peptide fragments into insoluble fibrils. During fibril formation, soluble proteins with dominating α-helices are converted into insoluble proteins with preferred β-sheets [348,349]. This conversion is favored by acidic pH values, as found in endosomes. The pathogenesis of Alzheimer’s disease is linked to the presence of very low pH values in endosomes [350]. This hyper-acidification enhances the formation of amyloid β fibrils and also affects the clearance of these fibrils.

Postmortem measurements revealed a slight decrease in pH values in the brain and cerebrospinal fluid of humans and mice with an increasing age [351]. In Alzheimer’s patients, the pH values of brain and cerebrospinal fluid were lower compared to controls without this pathology [351,352]. Moreover, lower brain pH values were associated with a more severe course of the disease [352].

Disturbances in iron metabolism have also been discussed in the pathogenesis of neurodegenerative disorders. Enhanced values of brain iron are observed in several disorders, which are collectively categorized as neurodegeneration with brain iron accumulation [353]. Iron accumulation within mitochondria contributes to oxidative stress, mitochondrial dysfunction, and neurodegeneration in Alzheimer’s disease, Parkinson’s disease, and others [354]. Unbalanced iron ions are able to oxidize lipids and other cell constituents, which leads, in consequence, to cell death by ferroptosis [355].

In Morbus Wilson, the excessive accumulation of copper ions in the liver, brain, and other organs is associated with serious damaging reactions [356]. In Alzheimer’s disease, increased values of free copper have been observed, which are not bound to ceruloplasmin [357,358].

Dysfunctions in heme metabolism are another source for neurotoxic events. In intracerebral and subarachnoid hemorrhages, the powerful oxidant free heme can result from hemoglobin released from defective red blood cells [359,360] after the exhaustion of the protective proteins haptoglobin and hemopexin [222,223,225]. The flat and hydrophobic free heme inserts easily into biological membranes, lipoproteins, and hydrophobic areas of proteins. At these loci, it favors oxidative processes and, thus, contributes to the development of atherosclerotic lesions [300], lipid peroxidation [301], neurodegenerative processes [360], and the induction of inflammatory events [32,361]. Amyloid β forms a complex with free heme, which promotes the formation of amyloid β fibrils [362]. Enhanced levels of haptoglobin in the sera of Alzheimer’s patients versus controls support the hypothesis that neuroinflammation and heme-induced stress contribute to the pathogenesis of Alzheimer’s disease [363]. Contrariwise, the hemopexin level was lower in the cerebrospinal fluid in patients with Alzheimer’s disease [364]. The different behavior between haptoglobin and hemopexin is maybe caused by the fact that human haptoglobin is an acute-phase protein, unlike human hemopexin [365,366].

## 5. Conclusions

To better understand the miraculous facets of the aging process, I prefer a thermodynamic explanation of aging, resulting in an increased energy deficit with an increasing age [2,3]. This hypothesis is based on the known fact that chemical and physical impacts permanently disturb the high order of biomolecules, cell organelles, cells, and tissues in organisms. Besides the existence of protective systems and repair mechanisms, a high growth rate significantly contributes to combat, with disturbing impacts during infantile and juvenile age. After an organism reaches its optimal size, growth effects play only a minor role as protective mechanism. As a result, physiological impairments develop, first of all slowly, but become more pronounced at advanced age.

Of course, in describing the aging process, many scientists have focused their attention on the expression of age-dependent biomarkers such as epigenetic alterations, telomere attrition, and factors playing a role in human longevity. According to the aforementioned hypothesis about aging, I directed the main focus here to potential age-dependent alterations in cellular ATP production and the (patho)physiological consequences derived from these deviations in energy metabolism. Some major consequences are the appearance of hypoxic states and dysfunctional mitochondria, alterations in cellular redox homeostasis, and the development of low-grade persistent inflammation. Host-derived cytotoxic agents play a significant role in the development of these restrictive impacts.

The human organism is equipped with powerful protective systems to resist a wide variety of external and internal impacts, to repair any damage, and to restore and maintain cell and tissue homeostasis. In many physiological processes, during the activation of immune cells and under stress situations, an enhanced formation of host-derived cytotoxic agents can occur. To avoid any damage by these agents, ready-to-use antagonizing principles exist, which immediately inactivate and eliminate these agents [8]. Serious problems arise when cytotoxic agents are very strongly expressed and act over a long time. Under these conditions, protection is limited, and cells and tissues are progressively damaged.

In this review, I applied the concept of host-derived cytotoxic agents and their interplay with antagonizing principles to the aging process and directed the focus to oxidant-based cytotoxic agents. During aging, there is a gradual decline in the production of energy equivalents and the supply of tissues with dioxygen and nutrients. Against this background, a chronic inflammatory state develops in older persons. In this persistent inflammation state, known as inflammaging, activated immune cells and pro-inflammatory mediators are present and coincide with a state of immunosuppression. In turn, oxidant-based cytotoxic agents considerably affect the cells and tissue integrity in older individuals and contribute to the development of age-related diseases. The balance between cytotoxic agents and antagonizing principles is crucial for the further fate of inflammation. In stressed and hypoxic tissues, the up-regulation of cellular redox homeostasis components and anti-oxidative proteins occurs. These adaptations have, however, long-lasting consequences for the distribution of energy substrates in relation to other physiological processes. More and more energy substrates are needed to stabilize protective mechanisms. These energy substrates are lacking for other physiological processes like active movement, mental processes, sensing the environment, and creating reserves for immunological defense.

It is very hard to predict how the imbalance between cytotoxic agents and protective mechanisms will affect the health status in a given person during aging. As protection against different oxidants and other cytotoxic agents plays an important role in stabilizing immune responses and preventing disease progression, a thorough analysis of the status of protective mechanisms is highly mandatory for older individuals.

## Figures and Tables

**Figure 1 biomolecules-15-00547-f001:**
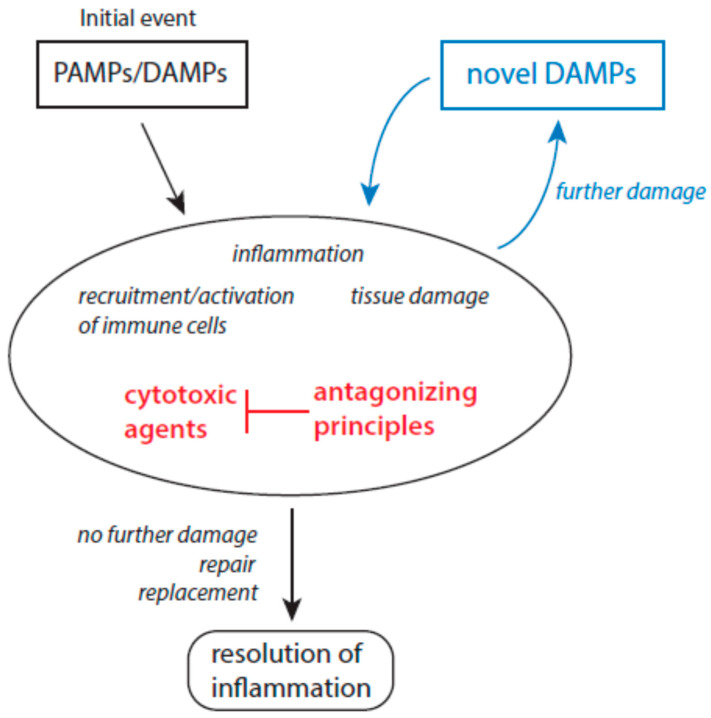
Roles of molecular patterns, tissue damage, and cytotoxic agents during inflammation. The further fate of inflammation depends on the individual state of protective mechanisms to antagonize cytotoxic agents. Resolution of inflammation is induced when no further damage occurs. As long as novel DAMPs are released from undergoing immune and tissue cells at inflammatory sites, the inflammation persists (presented in blue). Abbreviations: DAMPs—damage-associated molecular patterns and PAMPs—pathogen-associated molecular patterns.

**Figure 2 biomolecules-15-00547-f002:**
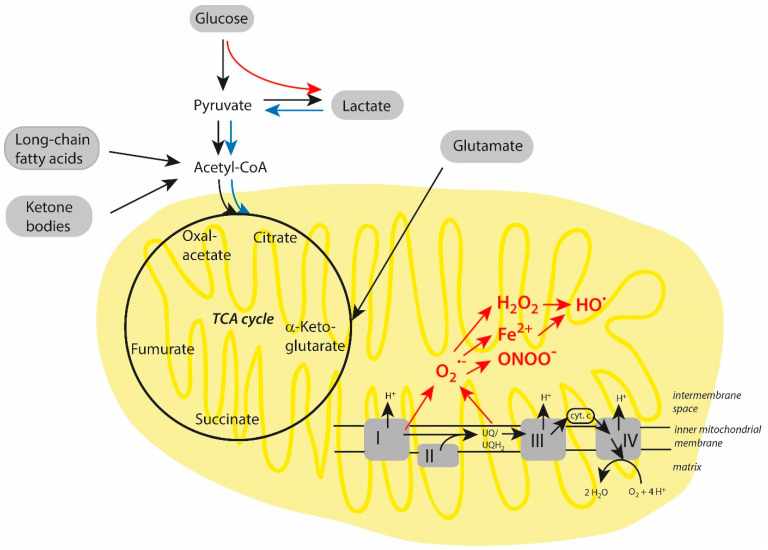
Possible alterations in the energy metabolism in stressed cells. Potential energy fuels are presented against grey background. Concerning amino acids as energy substrate, only glutamate is included. All other amino acids are also converted into intermediate products, which are applied at different places in the TCA cycle. In hypoxic cells, glycolysis is highly up-regulated, with lactate as final product (red arrow). Lactate can be used by neighboring cells as fuel to increase mitochondrial processes of energy production (blue arrows). In stressed cells, dysfunctional mitochondria are a source for enhanced production of oxidant-based cytotoxic agents (given in red), starting with superoxide anion radicals and followed by hydrogen peroxide, ferrous ions, hydroxyl radicals, and peroxynitrite. Further explanations are given in the text.

**Figure 3 biomolecules-15-00547-f003:**
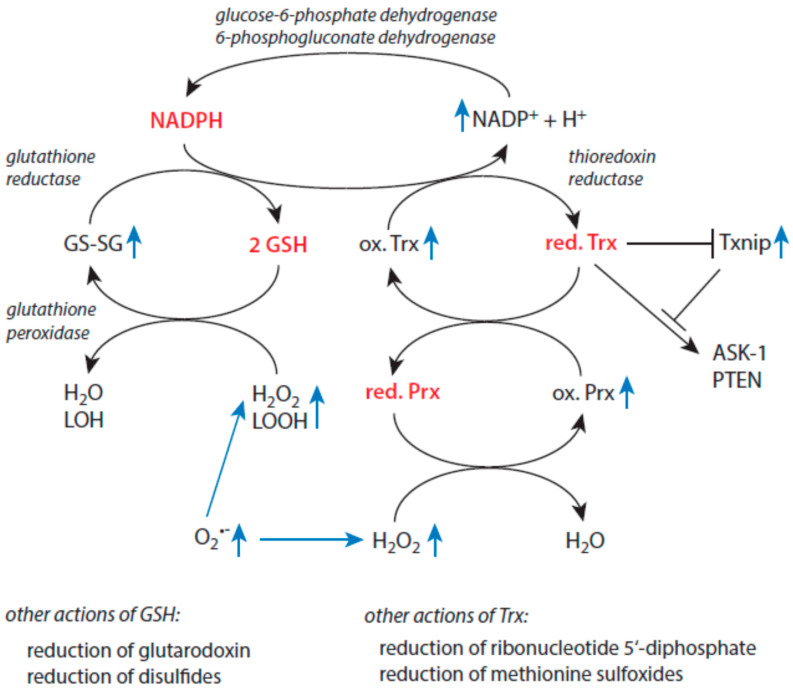
Basic mechanisms of redox regulation in cells. Metabolites that dominate in resting cells are given in red. The enhanced formation of reactive species like superoxide anion radicals and hydrogen peroxide under oxidative stress, as well as the following increases in other redox metabolites, are indicated by blue arrows. Abbreviations: ASK-1—apoptosis-signal-regulating kinase 1, GSH—glutathione, GS-SG—oxidized glutathione, LOH—lipid alcohol, LOOH—lipid hydroperoxide, Prx—peroxiredoxin, PTEN—phosphatase and tensin homolog, Trx—thioredoxin, and Txnip—thioredoxin-interacting protein.

**Table 1 biomolecules-15-00547-t001:** Key features and properties of the two main phases of acute inflammation.

Predominant Features or Properties	Initiation and Propagation of Inflammation	Resolution of Inflammation
Immune response	Recruitment and activation of immune cells	Deactivation of immune cells, immunosuppression
Presence of molecular patterns	Yes	No
Main physiological processes	Elimination of pathogens, removal of damaged cells and cell debris	Repair processes, synthesis of novel extracellular matrix
Typical mediators	IL-1, IL-6, IL-15, IL-8, TNF-α	TGF-β, IL-10, VEGF, lipoxins
Presence of CRP and SAA	Enhanced values	Decreasing values
Macrophage subtypes	M1 type	M2 type
Major pathways for ATP production in macrophages	Glycolysis	Oxidative phosphorylation
Presence of MDSCs	Low	Enhanced

**Table 2 biomolecules-15-00547-t002:** Neutrophil-derived cytotoxic agents and their antagonizing principles.

Cytotoxic Agent	Antagonizing Principles	References
Superoxide anion radicals	Superoxide dismutases	[58,59,60,61]
Hydrogen peroxide	Catalase, peroxiredoxins, glutathione peroxidases	[62,63,64]
Hydroxyl radicals	Limited protection by carbohydrates	[65]
Free transition metal ions	Proper control over all aspects of iron and copper ion metabolism	[66,67,68,69,70]
Myeloperoxidase	Ceruloplasmin	[71,72,73,74]
Hypochlorous acid, hypobromous acid	SCN^−^, taurine, glutathione, ascorbate	[75,76,77]
Hypothiocyanite	Glutathione	[78]
Elastase	α_1_-antitrypsin (A1AT), secretory leukocyte protease inhibitor (SLPI), elafin, serpin B1, α_2_-macroglobulin	[79,80,81,82,83]
Cathepsin G	A1AT, SLPI, α_1_-antichymotrypsin	[79,80,81,84,85,86]
Proteinase 3	A1AT, elafin	[79,80,82,83]

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
