# Peer review of "Oxidant-Based Cytotoxic Agents During Aging: From Disturbed Energy Metabolism to Chronic Inflammation and Disease Progression"

_biomolecules, 2025, doi:10.3390/biom15040547_

Round 1
Reviewer 1 Report
Comments and Suggestions for Authors
The manuscript represents a comprehensive and extremely detailed summary of the existing knowledge on cytotoxic agents involved in physiological alterations during aging. This is an original manuscript on a topic of interest in several areas of cell biology and immunology, especially in the field of oxidative stress. Overall, the review is clearly written and well structured. References are pertinent.
Due to the key features of aging, the title, in addition to oxidant-based cytotoxic agents, should include inflammation.
Author Response
Response to reviewer 1
Thank you very much for your review and your helpful comments. The following changes have been performed according to your suggestion.
Due to the key features of aging, the title, in addition to oxidant-based cytotoxic agents, should include inflammation.
Inflammation is now included in the title.
Reviewer 2 Report
Comments and Suggestions for Authors
Manuscript ID: Biomolecules-3540612
Manuscript Title: Oxidant-based cytotoxic agents during aging
Journal: Biomolecules
The article discusses how oxidant-based cytotoxins can cause cell and tissue damage over time. The authors also explore relevant mechanisms, including inflammation, mitochondrial dysfunction, and redox homeostasis, linking these factors to age-related diseases. However, the manuscript requires several revisions as follows:
- The title of the article refers Oxidant-based cytotoxic agents, but the article mentions several mechanisms such as inflammation and mitochondrial dysfunction which is confusing.
- What is “biological material”, please explain.
- Please reduce the number of keywords and focus on the key points.
- The abstract contains too much background information. Please focus more on the object of your research.
- Why discuss NLRP3 inflammation?
- In the conclusion section, please focus on the object of your work and highlight the research significance and innovation.
Comments on the Quality of English Language
English Language can be improved.
Author Response
Response to reviewer 2
Thank you very much for your review and your helpful comments. The following changes have been performed according to your suggestion.
The title of the article refers Oxidant-based cytotoxic agents, but the article mentions several mechanisms such as inflammation and mitochondrial dysfunction which is confusing.
The title has been changed.
What is “biological material”, please explain.
I replaced the term “biological material” by biomolecules and other terms.
Please reduce the number of keywords and focus on the key points.
The last two keywords have been omitted.
The abstract contains too much background information. Please focus more on the object of your research.
Abstract has been changed to better on the research object.
Why discuss NLRP3 inflammation?
In section 3.6, several mitochondria-derived DAMPs are mentioned. These DAMPs can contribute to NLRP3 inflammasome activation and to the release of pro-inflammatory cytokines. I focused on this fact, but I did not go into detail. This sentence has been included to demonstrate the complexity how dysfunctional mitochondria promote inflammatory processes.
In the conclusion section, please focus on the object of your work and highlight the research significance and innovation.
Conclusion has been extended to consider these aspects.
Reviewer 3 Report
Comments and Suggestions for Authors
This review provides valuable insights into the effects of oxidant-based cytotoxic agents and cellular responses during aging. The manuscript is well-organized and clearly written. While it thoroughly addresses key aspects of aging, particularly mitochondrial dysfunction and inflammaging, which are critical topics, some important biomarkers—such as epigenetic alterations and telomere attrition—are not covered. I suggest that the author briefly discuss these widely recognized aging biomarkers and explain why some of them are not addressed in detail in this review. Additionally, one minor suggestion is to enhance the appearance of the figures, particularly Figure 2, to make them more visually appealing. Overall, this is an excellent manuscript that will make a valuable contribution to the field of aging research.
Author Response
Response to reviewer 3
Thank you very much for your review and your helpful comments. The following changes have been performed according to your suggestion.
While it thoroughly addresses key aspects of aging, particularly mitochondrial dysfunction and inflammaging, which are critical topics, some important biomarkers—such as epigenetic alterations and telomere attrition—are not covered. I suggest that the author briefly discuss these widely recognized aging biomarkers and explain why some of them are not addressed in detail in this review.
Age-dependent biomarkers such as epigenetic alterations, telomere attrition, and factors playing a role in human longevity are shortly mentioned in the Conclusion section. In this context, I explained this main focus of this review.
Additionally, one minor suggestion is to enhance the appearance of the figures, particularly Figure 2, to make them more visually appealing.
A new version of figure 2 is included.
Reviewer 4 Report
Comments and Suggestions for Authors
The compilation is well handled and described in detail. I would be grateful if you could answer my questions in the text. I have also made some corrections in the PDF file. You should also take those parts into consideration.
As a person who has been studying neuroimmunopathogenesis and molecular pathogenesis in inflammatory processes for many years, if I make a general evaluation,
a) There is confusion in some definitions. It will be more understandable if those parts are corrected. (I have specified in detail)
b) I have seen missing personal comments under some special sections. Your opinion and contribution should be added in detail. After all, if you are writing this compilation, you have a good background. Your personal thoughts will add value to the compilation.
I strongly recommend that you make an overall assessment of the conclusions and implications of the points discussed.

Author Response
Response to reviewer 4
Thank you very much for your review and your helpful comments. The following changes have been performed according to your suggestion.
I have also made some corrections in the PDF file. You should also take those parts into consideration.
- a) There is confusion in some definitions. It will be more understandable if those parts are corrected. (I have specified in detail)
Yes, I considered your helpful remarks and performed changes accordingly.
- b) I have seen missing personal comments under some special sections. Your opinion and contribution should be added in detail. After all, if you are writing this compilation, you have a good background. Your personal thoughts will add value to the compilation.
I strongly recommend that you make an overall assessment of the conclusions and implications of the points discussed.
I considerably extended the Conclusion section to consider your suggestions. Abstract is also slightly changed.
Round 2
Reviewer 2 Report
Comments and Suggestions for Authors
The author carefully revised the manuscript and the quality of the manuscript was improved. It is recommend to accept the manuscript.